# A New Approach for Timing Post-Emergence Weed Control Measures in Crops: The Use of the Differential Form of the Emergence Model

Jordi Izquierdo [1],*[ID], Clara Prats [2][ID], Montserrat Gallart [1] and Daniel López [2]

1   Department of Agri-Food Engineering and Biotechnology, Universitat Politècnica de Catalunya, 08860 Castelldefels, Spain
2   Department of Physics, Universitat Politècnica de Catalunya, 08860 Castelldefels, Spain
*   Correspondence: jordi.izquierdo@upc.edu; Tel.: +34-93-5521084

**Abstract:** Models based on thermal or hydrothermal time are used to predict the seedling emergence pattern of weeds. These models rely on sigmoidal functions such as Gompertz, Weibull or logistic, in which daily soil temperature and moisture data are inputs and the percentage of total expected emergences is the output. The models give good predictions at local and regional scales but they lose accuracy when extrapolated to different geographic areas from where the equations were developed. They also must be validated prior to their release and have subjectivity of the date to start the accumulation of the degree-days. We propose the use of the differential form of the function rather than the integrated form. Under this approach, the starting date to accumulate degree-days is set to the week before the first weed emergence is recorded (if recorded on a weekly basis) and emergence predictions only rely on the current sigmoidal relationship between data recordings. When the weed emergence rate in the field decreases, the relationship between data recordings and time, measured either as thermal or hydrothermal degrees, starts to decrease. When the derivative of the emergence over time falls below a threshold that should be set up based on our knowledge of the economic threshold of the species, a post-emergence weed control measure should be carried out. Under this approach, weekly counts of weeds must be recorded until the derivative reaches the threshold. This approach has been checked on 39 data sets of different weeds in different crops and seasons by applying the differential form of the Gompertz function, obtaining a correlation of 0.99 between the predicted and the observed emergence. The methodology could be particularly useful when timing control measures in cropping areas with unknown or very little knowledge of the species and their emergence pattern.

**Keywords:** Gompertz model; weed management; seedling emergence; model construction; hydrothermal time

## 1. Introduction

In cereal systems, weed control is crucial for the crop to achieve the potential yield of modern varieties. Most weed management programs focus on eliminating seedlings as this is the stage at which the weed is more sensitive to control measures [1]. Although weed seedlings are expected to emerge for a period of time after crop sowing, environmental conditions may delay or advance the onset of emergence [2]. Crop and weed interactions are highly dependent on environmental factors, cultural practices, and the crop and weed involved. Precise knowledge of the time span in which weed seedlings emerge would help farmers to optimize the timing and maximize the efficacy of post-emergence weed control measures such as mechanical weeding or herbicide spraying. This is becoming increasingly relevant for growers because of the current pressure to reduce chemical input or to adopt non-chemical methods [3].

In order to fulfill these needs, mechanistic and empirical approaches to predict weed seedling emergence timing and density have been developed for single species. Mechanistic models are based on biophysical processes and require a thorough knowledge of the biology and ecology of the species. Because of its complexity, they provide a robust approach to predicting the effect of the date of soil cultivation on the emergence of the weed species [4] and also the effect of different cropping systems [5,6]. However, they require parameters that may be difficult to measure and consequently they do not have the simplicity and flexibility required as a practical decision-support system for timing weed control measures [7]. Empirical models are based on seedling emergence observations related to soil moisture and temperature measurements. Once these models are developed and validated, they can provide accurate predictions of weed emergence provided that the fields are located in the same or similar region and/or had similar weather conditions. There are two types of empirical models; thermal time models, which solely use soil temperature to describe emergence [8–10] and hydrothermal time models, which include soil moisture in the calculations [11–13]. Hydrothermal time models are expected to give more accurate information about emergence than thermal time, particularly in dryland ecosystems in which seedling emergence is highly reliant on rainfall.

The time course of cumulative emergence of weed seedlings in the field follows the kinetics of a sigmoid-shaped curve [14]. Consequently, empirical seedling emergence models are mostly based on non-linear regression of sigmoidal-shape functions such as Gompertz, Weibull, or logistic in its integrated form, that relate thermal or hydrothermal time with cumulative emergences. These models are developed using data from several fields and, after being validated in different locations, they provide good predictions of weed emergence at the local and even at the regional scale [15,16]. When these models are applied to fields located in different geographic areas and/or when weed populations differ genetically or in dormancy status, they may lose accuracy [17]. As Grundy [7] pointed out, the complexity of the interactions between weather variables and seed status can sometimes produce considerable under or over-estimation of the weed emergence. In these situations, the models have to be reparametrized which implies more years of emergence observations in fields [18]. Hardly any single regression function in its integrated form has been able to describe the emergence pattern of all weeds or the same weed species in all possible scenarios. Some authors have attempted a non-parametric approach [19], which is more flexible but it resembles a black box with little biological underpinnings and is prone to overfitting [20]. The subjectivity of the date to begin the accumulation of degree-days is another limitation of these models. As González-Andújar et al. [21] reported, the accumulation of degree-days in these studies started at different times in the season such as the last soil tillage event or burndown herbicide application, crop sowing or an arbitrary date such as 1 January for summer-growing weeds in cold climates of the northern Hemisphere or winter-growing weeds in warm climates of the Southern Hemisphere. However, these dates have no biological or ecological basis, as seeds are not monitored for accumulation of thermal or hydrothermal time since they ripen, consequently, the moment at which they lose primary dormancy and start progressing towards germination is not known and it will probably differ from the dates mentioned above.

Although non-linear regression is the most used empirical approach to model weed emergence in fields, other approaches have also been applied. Soft-computing techniques such as genetic algorithms [22,23], artificial neural networks [24] or survival analyses [25] have been demonstrated to be good descriptors of germination in lab-based tests or field emergence studies, but they also have to be parameterized before being used as a predictive tool. A description of these techniques can be found in González-Andújar et al. [21].

We present here a new approach when using non-linear regression functions to predict weed emergence in the field: the use of the differential form of the function. The differential form of a given function defines the relationships between a physical quantity and its rate of change. The relationship is built by iteratively adding new data into the function. The more data, the more accurate the curve will be. Under this approach, the starting

point is the first occurrence of the variable that is being recorded (i.e., the date of the first emerged seedling). A validation of the parameters of the function is no longer required as the parameters are estimated based on the current relationship between data, which is sigmoidal-shape-like. The value of the parameters will reflect the underlying relationships between the variables and the factors of the process being modeled (i.e., the parameters will reflect the relationships between the emergence and the current season weather and characteristics of the seeds and soil).

Our hypothesis is that the use of the differential form of a non-linear regression function instead of its integrated form avoids the issues discussed above, allowing higher flexibility to the model while retaining the meaningfulness and understanding of the biology of the parameters associated with seedling emergence. Our objective was to test the feasibility and accuracy of the differential form of a function as a substitute for the integrated form for predicting the timing of a weed control measure in those situations in which the integrated form of the model may not work accurately. Specifically, we checked whether the differential approach predicts the final number of weed seedlings that are expected to emerge in the field and the week in which such quantity is reached. Within this framework, we also checked whether smoothing count data prior to the implementation of the methodology can improve the accuracy of these predictions.

## 2. Materials and Methods

In order to test the feasibility of the differential form of a function to predict weed emergence, any sigmoid-shaped curve that was able to describe the emergence pattern of a weed can be selected. Among all possible functions (logistic, Weibull, Gompertz, ... ), we chose the Gompertz function for the demonstration because in previous studies we observed that the data sets already collected that we will use for this purpose had been successfully fitted to the integral form of this function, showing residual mean squares of the fitting with lower values compared to the other non-linear functions [9,26,27]. However, any other sigmoid-type function could be used as well.

By using the Gompertz function, the pattern of the cumulative seedling emergence of the plants is related to time according to:

$$E = K \exp^{-Ae^{-\mu_m t_H}} \tag{1}$$

where $E$ is the cumulative seedling emergence measured as seedlings m$^{-2}$, $K$ is the asymptote of the function, that is, the maximum number of seedlings that will emerge for the species in that season, the parameter $A$ is defined as $A = \ln \frac{K}{E_0}$ with $E_o$ being the initial number of seedlings per m$^{-2}$, $\mu_m$ is a constant related with the emergence rate of increase and $t_H$, is time measured as a hydrothermal time in MPa °C day$^{-1}$ [28].

The differential form of this function is:

$$\frac{dE}{dt_H} = \mu_m A \, e^{-\mu_m t_H} \tag{2}$$

which can be also expressed as:

$$\frac{1}{E} \frac{dE}{dt_H} = \mu_m lnK - \mu_m lnE \tag{3}$$

where $\frac{dE}{dt_H}$ is the emergence speed, $\frac{1}{E} \frac{dE}{dt_H}$ is the specific emergence speed which shows the rate of increase of the emergence. The advantage of rewriting the differential form as shown in Equation (3) is that it resembles the equation of a linear model, so that facilitating the parameterization process. If we plot $\frac{1}{E} \frac{dE}{dt_H}$ versus $lnE$, we have a decreasing linear relationship between both terms in which $\mu_m$ is given by the slope of the line and $K$ is obtained from the intercept term as $K = e^{\frac{a}{\mu_m}}$ being $a$ the x-intercept of the linear relationship.

## *2.1. Experimental Data*

The differential form of the Gompertz function was tested on 39 emergence data sets of several weed species infesting crops that were obtained in previous studies in Spain and Portugal that can be found in Dorado et al. [9], Izquierdo et al. [26,27] or unpublished (Table 1). In all cases, the emergence of these weed species over time on a thermal or hydrothermal scale was successfully described by using the integrated form of the Gompertz function. The data came from a variety of weeds, crops, and crop management. Some of them are from rainfed winter cereals crops such as wheat and barley, others from summer irrigated crops such as corn, and a few more from perennial irrigated citrus orchards in which the weed emergence was monitored in spring. For each weed species, the emergence was counted weekly in 20 randomly placed permanent quadrats of 30 cm $\times$ 40 cm to 50 cm $\times$ 50 cm depending on the field. After being counted, weed seedlings were removed from the soil with minimum soil disturbance. Data loggers were placed in each field to record soil temperature hourly throughout the growing season and daily means were used in the assessment of the accumulated thermal or hydrothermal time for the species. When emergence was related to hydrothermal time, daily soil moisture was estimated using rainfall and evaporation data from weather stations and the soil texture. The beginning of the time accumulation was set at crop sowing for cereals or soil tillage for citrus as these dates are easily remembered by farmers and the soil disturbance due to the tillage has a stimulatory effect on the germination of non-dormant seeds by the sudden exposure of the seed to light and oxygen [29]. In all data sets, the observed accumulated emergence of the weed was described by the integrated form of the Gompertz function. For more details about the specific methodology used to collect and analyze the data, see the references mentioned above and in Table 1.

**Table 1.** Data sets used from Spain and Portugal ($^+$) sites to test the differential form of the Gompertz function. More details are in the supplementary material (I).

| Weed | Crop | Sites | Year | Water Regime | Source * |
|---|---|---|---|---|---|
| *Digitaria sanguinalis* (L.) Scop. | Citrus | Huelva1 | 2008 | Irrigated | 4 |
| | | Huelva2 | 2008 | Irrigated | 4 |
| | Corn | Arganda | 2005, 2006 | Irrigated | 1 |
| | | Golega + | 2007 | Irrigated | 1 |
| | | La Roca1 | 2007 | Irrigated | 4 |
| | | Miralcamp | 2010 | Irrigated | 4 |
| | | Mollerussa | 2010 | Irrigated | 4 |
| *Echinochloa crus-galli* (L.) P. Beauv. | Corn | Golega + | 2006 | Irrigated | 1 |
| | | La Roca1 | 2006, 2007 | Irrigated | 4 |
| | | La Roca2 | 2008 | Irrigated | 4 |
| | | La Roca3 | 2009 | Irrigated | 4 |
| | | Miralcamp | 2010 | Irrigated | 4 |
| | | Mollerussa | 2010 | Irrigated | 4 |
| *Lolium rigidum* Gaudin | Cereal † | Albacete | 2007, 2008 | Dryland | 4 |
| | | Calaf | 2006, 2007, 2008 | Dryland | 3 |
| | | El Encín | 2008 | Dryland | 3 |
| | | Huelva | 2006 | Dryland | 4 |
| | | Igualada | 2006, 2007, 2008 | Dryland | 3 |
| | | Murillo | 2008 | Dryland | 3 |

**Table 1.** *Cont.*

| Weed | Crop | Sites | Year | Water Regime | Source * |
|------|------|-------|------|--------------|----------|
| *Papaver rhoeas* L. | Cereal † | Calaf | 2006, 2007, 2008 | Dryland | 2 |
| | | El Encín | 2008 | Dryland | 2 |
| | | Igualada | 2006, 2007, 2008 | Dryland | 2 |
| | | Murillo | 2008 | Dryland | 2 |
| *Phalaris brachystachis* Link | Cereal † | Huelva | 2008 | Dryland | 4 |
| *Phalaris paradoxa* L. | Cereal † | Tajonar | 2008 | Dryland | 4 |
| *Portulaca oleracea* L. | Citrus | Huelva1 | 2008 | Irrigated | 4 |
| | | Huelva2 | 2008 | Irrigated | 4 |
| | Corn | La Roca1 | 2006 | Irrigated | 4 |

* 1: Dorado et al. (2009) [26]; 2: Izquierdo et al. (2009) [9]; 3: Izquierdo et al. (2013) [27]; 4: Unpublished results.
† Winter cereal.

### 2.2. Fitting the Differential Form of the Gompertz Function

When fitting the differential form of a function to the emergence data over time, the calculation starts when the first emergence is recorded. If field monitoring is conducted on at weekly basis, the accumulated time that is attributed to the first emergence is the thermal or hydrothermal time accumulated during the week before. In order to forecast the number of weed seedlings, Equation (3) is fitted every time weed emergence is counted. Any new measurement is added and new predictions are conducted.

In order to better see patterns or trends in time-series and improve the accuracy of forecasts when modeling, the application of algorithms (called filters from now on) is very common. The filters remove the short-term irregularities caused by the effect of the random variation. When fitting the integrated form of the emergence models, the resulting model is already a smooth representation of the data because the parameters are estimated by statistical procedures that give the best fit for a set of data points. However, when applying the differential approach, we are using the original data points (i.e., direct counts of plants) for the calculations that have a potential random variability which may mask the trend and weaken the accuracy of the forecasts. The application of filters or smoothing techniques to direct counts may improve predictions. In order to see the effect of the smoothing on the accuracy of the predictions, we applied the differential approach without filters (O1) and smoothed them with a filter (O2).

A.　O2: Smoothing weed emergence data ($Ei$).

The filter applied to the counts is a three-point weighted numerical filter [30]. It is applied to the weed emergence experimental data points ($E_i$, $i$ = 2: $N - 1$) and we obtain a smoother data set [$\overline{E}_i$, $i$ = 2: $N - 1$].

$$\overline{E}_i = \frac{E_{i-1} \frac{t_{H,i+1}-t_{H,i}}{t_{H,i+1}-t_{H,i-1}} + E_i + E_{i+1} \frac{t_{H,i}-t_{H,i-1}}{t_{H,i+1}-t_{H,i-1}}}{3} \tag{4}$$

The new smoothed data set is composed of $N - 2$ pairs of values ($t_{H,i}$ and $\overline{E}_i$, $i$ = 2: $N - 1$). Recall that in O1 we maintain the original $N$ pair of emergence data values ($t_{H,i}$ and $E_i$).

B.　Estimating the numerical derivative of the raw (in O1) or smoothed (in O2) emergence

The numerical derivative of a certain point is estimated by assessing the difference between next ($i + 1$) and previous ($i - 1$) points [31], as shown in the following equation:

$$\frac{d\overline{E}_i}{dt_{H,i}} \cong \frac{\overline{E}_{i+1} - \overline{E}_{i-1}}{t_{H,i+1} - t_{H,i-1}} \tag{5}$$

This will provide a set of $N - 2$ (in O1) or $N - 4$ (in O2) numerical derivatives' values ($\frac{d\overline{E}_i}{dt_{H,i}}$) that will correspond to the following time points:

$$\overline{t}_{H,i} = \frac{t_{H,i+1} - t_{H,i-1}}{2} \tag{6}$$

C.　Smoothing the estimated derivatives

The set of estimated derivatives from step B were submitted again to a three-point weighted numerical filter (Equation (4)) so that possible excess of noise can be further eliminated. As a result, a $N - 4$ (in O1) or $N - 6$ (in O2) smoother set of derivatives ($\frac{d\overline{E}_i}{dt_{H,i}}$) at the time points $t'_i$ was obtained ($i = 3$: $N - 3$).

D.　Fitting Gompertz function

The emergence ($E_i$) or smoothed emergence ($E'_i$) and the smoothed derivatives (step C) were used to calculate the specific emergence rates (O1: $1/E_i \times dE_i/dt_i$; O2: $1/E'_i \times dE'_i/dt_i$) and the natural logarithms of emergence (O1: $ln\ E_i$; O2: $ln\ E'_i$) at each point $i$. Then, these pairs of values were used to fit a linear regression that has the following form:

$$\frac{1}{E}\frac{dE}{dt} = n - m \cdot \ln E \tag{7}$$

According to Equation (3), the parameters $n$ and $m$ can serve to determine the maximum specific emergence rate ($\mu_m$), being $\mu_m = m$, and the maximum expected seedling population of the field ($K$), being:

$$K = e^{n/m} \tag{8}$$

E.　Determination of the percentage of emergence

The proposed iterative fitting allows the assessment of the asymptote or maximum seedling population ($K$; plants m$^{-2}$). The higher the number of seedlings counts, the more accurate the estimation of the maximum seedling population. Given the number of seedlings at a certain moment ($E_i$), the percentage of the current emergence ($P_E$) relative to the expected final emergence ($K$) for that species and season can be determined directly through:

$$P_E = \frac{E_i}{K} \cdot 100 \tag{9}$$

F.　Implementation of the protocol

Although the protocol consists of 5 steps that are repeated each time that a new count is added to the experimental dataset, the mathematical operations involved are very simple. They can be implemented in any software that incorporates linear fitting. In our case, MatLab 8.0 was the statistical software used to automate the protocol and apply it to the different datasets.

Weed seedling emergence is characterized by an initial exponential growth that is followed by a negative acceleration phase until it reaches a zero-emergence rate and no more seedlings are counted. Due to this negative acceleration phase, the derivative of the emergence decreases every week until it reaches zero. The iterative procedure that we have described above can be stopped (and recording emergence in the field can be ceased) when the derivative decreases below a certain value or threshold that should correspond with a date in which any further weed seedling emergence will not significantly affect the yield of the crop. This threshold value, which represents a percentage of the total final emergence expected, must be set according to the economics of the weed-crop competition, the usual environmental conditions, and the weed and crop species involved. This knowledge must come from the literature (see review in Swanton et al. [32]) or the farmer or adviser's knowledge of the crop-weed interaction. Ultimately, the procedure (and the counting) can be stopped when the observed emergence reaches the predicted final emergence.

G.    Validation of the approach

Farmers accept that a weed control measure is effective when at least 95% of the weeds are controlled. Assuming in advance that any control measure properly conducted must control all weeds, we simply need to know when the field will have 95% of emerged weeds in that season because this date will be the time to apply the control measure. In order to achieve this date, the differential approach must give an accurate prediction of the proportion of seedlings that emerge over the expected total. As the differential model does not give us percentages but emergence rates $\frac{dE_i}{dt_{H,i}}$, we have to find which rate is equivalent to a 95% of emergence. For this purpose, we run the model with all data sets and checked if the rates of 0.03 (criterion A1) and 0.05 (criterion A2) plants (m$^2$ day)$^{-1}$ corresponded to an emergence of 95% and which one was the most accurate. The accuracy was checked by correlation. The emergences estimated with our approach were compared by correlation to both the observed emergence in the field and the expected emergence if we apply the integrated model. Additionally, each pair of values (predicted versus observed) were compared per field. If the value of the predicted emergence was within the observed emergence $\pm 10\%$ range, we considered it to have a good prediction; if it was within the observed value $\pm 20\%$, then we considered it to have a regular prediction.

Not only the predicted weed emergence for the season was checked. As this type of model is also used to estimate the best time to control the weed, the predicted week at which the 95% is reached was also validated by checking if this week matched the week observed in the field $\pm$ one week, which means if matches a 3-week interval. By adding $\pm$ one week we consider that during this period the efficacy of the control measure would not change significantly, based on the fact that the application time of most herbicides comprise several phenological stages.

In summary, in this study we tested which of the following criteria gives the best result to forecast the advent of 95% of weed emergence in the field:

A. Filtering both emergence and derivative data and (A1) ceasing counts when derivative drops below 0.03 plants (m$^2$ day)$^{-1}$ or (A2) ceasing counts when derivative drops below 0.05 plants (m$^2$ day)$^{-1}$.

B. Filtering only derivative data and ceasing counts when derivative drops below 0.05 plants (m$^2$ day)$^{-1}$. We did not check the threshold 0.03 plants (m$^2$ day)$^{-1}$ because we observed after applying A1 that the predictions with the threshold 0.03 plants (m$^2$ day)$^{-1}$ were always equal or slightly worse than with the threshold of 0.05 plants (m$^2$ day)$^{-1}$.

## 3. Results

### 3.1. Implementation of the Protocol in One Data Set: How It Works

As an example of the application of the differential approach, the data set of the emergence of *Papaver rhoeas* at Igualada in 2008 has been selected. A graphical display of the output that is obtained after applying the differential approach to every weekly count using two filters (criterion A) is shown in Figure 1, Table 2 and supplementary material (II). The first data for the linear regression described in step E of the methodology, the emergence rate, is obtained at the end of week 6. At week 7 we have the second emergence rate (Table 2A) and, consequently, both values can be related by linear regression (Figure 1). Based on this regression and according to Equation (8), the expected maximum emergence in the field would be $K$ = 204 plants m$^{-2}$. This estimation varies over the following three weeks as more data are being used in the calculations ($K$ = 189, 190, 192 plants m$^{-2}$ for each following week). At week 11 the numerical derivative of the emergence dropped below the threshold of 0.03 (criterion A1) and even below 0.05 plants (m$^2$ day)$^{-1}$ (criterion A2) (Table 2A). Consequently, field counts can be ceased. In this week 11, the predicted maximum emergence in the field was 193 plants m$^{-2}$ and the observed emergence was very similar, 191 plants m$^{-2}$. If we applied the integrated form of the Gompertz function to this data, we would have a predicted total emergence of 191 plants m$^{-2}$, in agreement with what was observed. However, it is important to point out that the observed emergence in the field remained steady at 191 plants m$^{-2}$ since week 9 (Table 2A), indicating that by

using the differential approach the maximum emergence was determined with a delay of two weeks. According to our criteria of considering a delay of only one week acceptable, the maximum emergence for this weed in this location was not accurately predicted by applying the differential approach with two filters. If we had applied one filter (criterion B), the maximum emergence would be predicted at week 9, which would have coincided with the week in which the maximum emergence was observed in the field (Table 2B). In consequence, we conclude that for this particular dataset criterion B of only filtering the derivative would have been more accurate.

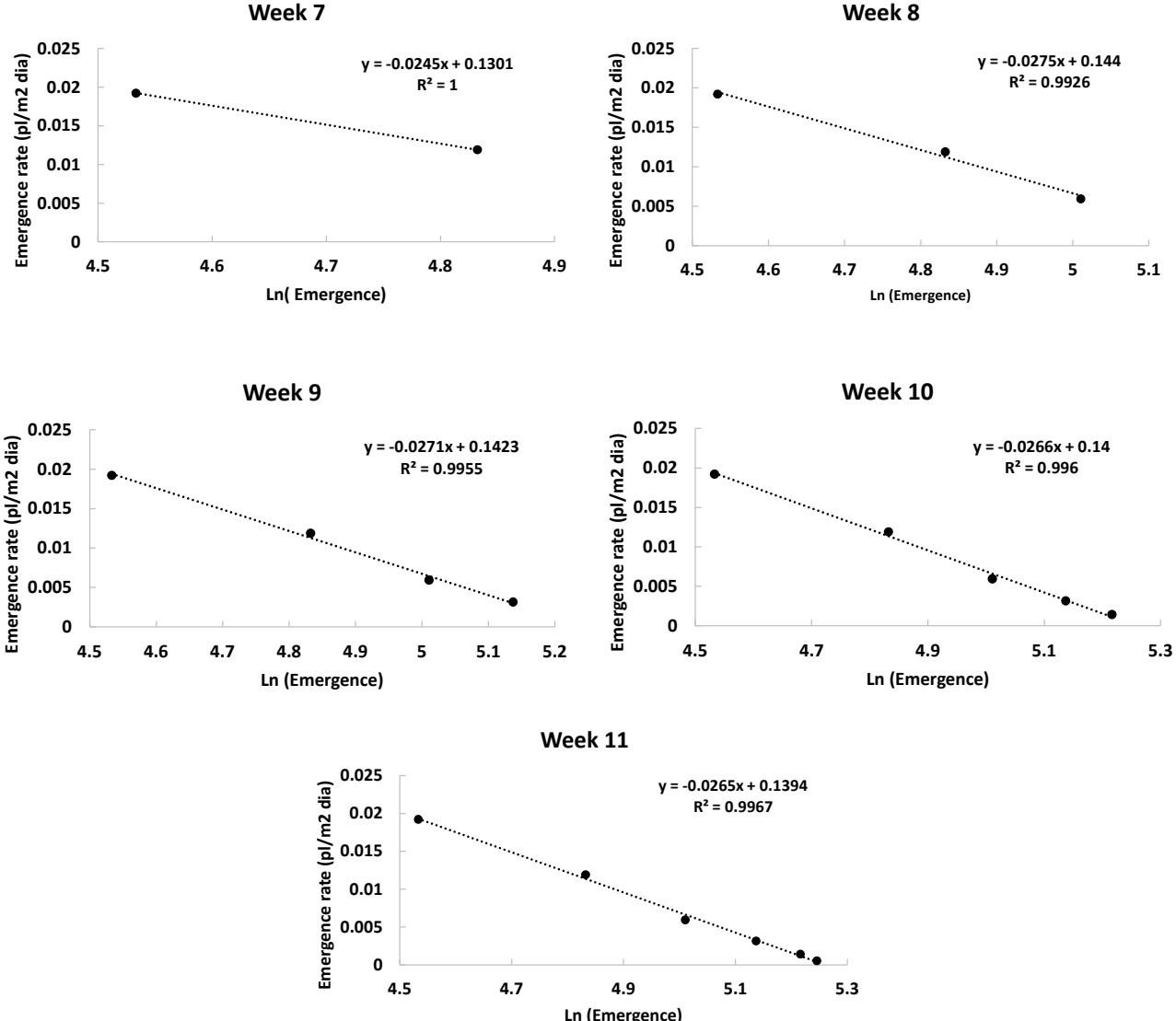

**Figure 1.** Example of graphical outputs of the application of the differential approach (see steps A–F in text) to the dataset *Papaver rhoeas*–Igualada-2008 to the data available each week using two filters. Numerical filters were applied to smooth both the emergence and derivative data. Weed seedling counts ceased at week 11 when the derivative was lower than 0.03 plants $(m^2 \ day)^{-1}$ (Table 2). The graphs show the linear regressions $(A = -mB + n)$ between the specific emergence rate $(\frac{1}{E_i} \frac{dE_i}{dt_{H,i}} = A)$ and the natural logarithm of the smoothed emergence ($\ln \overline{E}_i = B$). The regressions were calculated weekly, based on the emergences observed in the field ($E_i$) and the accumulated hydrothermal time ($t_{H,i}$). The maximum rate of emergence ($\mu_m$; $\mu_m = m$) and the maximum weed seedling population for the year ($K$; $K = e^{\frac{n}{m}}$) are estimated from the regression equation.

**Table 2.** Parameters of the graphical outputs of Figure 1 for the dataset *Papaver rhoeas*–Igualada-2008. Number of points involved in the linear regression, numerical derivative of the emergence with standard error, expected maximum weed seedling population according to the differential approach (*K* differential) and observed weed seedling population in the field (*E* experimental) when: (**A**) the emergence and the numerical derivative of the emergence were smoothed and (**B**) only the numerical derivative of the emergence was smoothed.

| A. Filter Applied to Smooth Emergence and Numerical Derivative | | | | | |
|---|---|---|---|---|---|
| Parameters | Week | | | | |
| | 7 | 8 | 9 | 10 | 11 |
| Data points of regression | 2 | 3 | 4 | 5 | 6 |
| Numerical derivative (pl.MPa/m$^2$.day) | 0.898 | 0.485 | 0.230 | 0.070 | 0.010 |
| *K* differential (pl/m$^2$) | 204 [NA-NA] | 189 [85–293] | 190 [131–250] | 192 [145–240] | 193 [155–231] |
| *E* experimental (pl/m$^2$) | 188 | 190 | 191 | 191 | 191 |
| B. Filter Applied to Smooth Numerical Derivative | | | | | |
| Parameters | Week | | | | |
| | 5 | 6 | 7 | 8 | 9 |
| Data points of regression | 2 | 3 | 4 | 5 | 6 |
| Numerical derivative (pl.MPa/m$^2$.day) | 1.061 | 0.941 | 0.571 | 0.183 | 0.030 |
| *K* differential (pl/m$^2$) | 214 [NA-NA] | 200 [136–263] | 189 [124–254] | 192 [141–244] | 194 [150–238] |
| *E* experimental (pl/m$^2$) | 148 | 174 | 188 | 190 | 191 |

### 3.2. Implementation of the Protocol in All Data Sets and Accuracy Obtained

If we consider the 39 datasets, the accuracy in determining the week of the maximum weed seedling emergence in the field was greater when criterion B was applied (81% of cases) instead of criterion A2 (54%) or criterion A1 (43%). Criterion B could not be applied to three datasets because the data did not drop below the thresholds. In the other seven datasets, the predicted week of maximum emergence differed by more than 2 weeks, being five of them predicted in advance. In four of this five, the mismatch was attributed to the low number of seedlings that emerged each week, which soon made the derivatives to be very small and fall below the threshold. These datasets had a total weed seedling population lower than 20 plants m$^{-2}$.

Regarding the accuracy in predicting the maximum seedling emergence (given by *K* differential), this figure was compared with both the observed total emergence in the field (*K* experimental) and the maximum emergence predicted by applying the integrated form (*K* integrated). The correlations between these values were always greater than 0.95, with the regression line slopes close to 1 indicating a good estimation (Figure 2). If we compare the estimated and the observed seedling emergence for every single field, we observed that the accuracy was greater when we applied criterion A1 (74% of cases with less than 10% difference) than when applying criterion A2 (73%) or criterion B (63%). If we increased the deviation up to $\pm20\%$, the percentage of datasets in which the maximum population is predicted with accuracy increased to 91% for criteria A1 and A2 and 86% for criterion B.

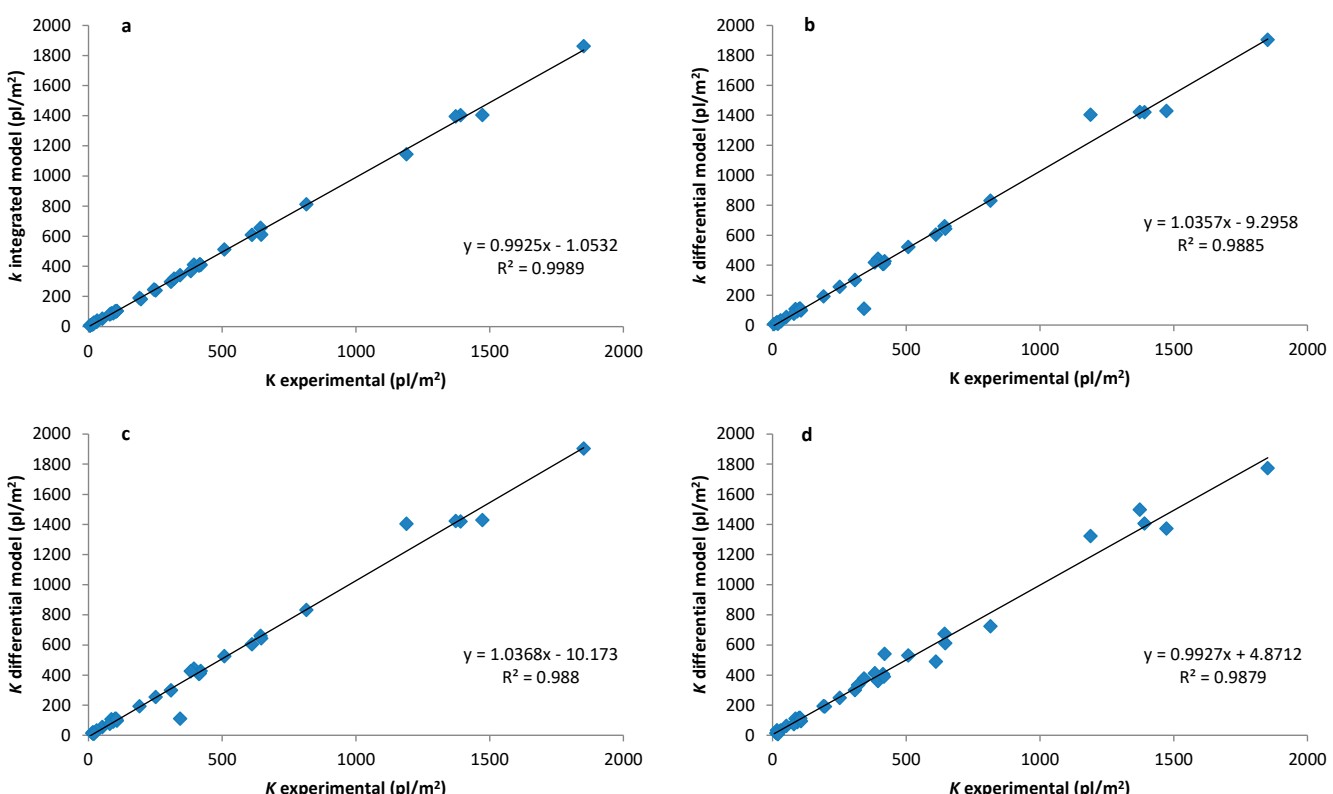

**Figure 2.** Relationships between the maximum weed seedling population observed in the field (*K* experimental) and the predicted weed seedling population according to different approaches: (**a**) the integrated approach (n = 39); (**b**) the differential approach with filters applied to the emergence and the derivative of the emergence and criteria to cease counts being derivative <0.03 (n = 35); (**c**) the differential approach with filters applied to the emergence and derivative of the emergence and criteria to cease counts being derivative <0.05 (n = 33) and (**d**) the differential approach with the filter applied to the derivative of emergence and criteria to cease counts being derivative <0.05 (n = 35).

## 4. Discussion

The differential approach was implemented successfully in most datasets. The application of two filters during the calculations instead of one filter increased the accuracy in predicting the maximum expected emergence because outliers were smoothed and fitting improved. However, the differential approach requires to have more counts (weeks with emergence data) to give estimations because we have to both calculate a derivative and use filters. In order to obtain an estimation of the emergence, data from three weeks in advance are needed (see Equations (3) and (4)). This three-week gap between the week in which counts are performed and the week for which the emergence can be estimated could be reduced to only two if we only apply a filter to the derivative. Another negative aspect of the application of two filters is that it makes that the first possible estimation of the expected seedling emergence is only obtained after seven weeks of counts (see Supplementary Material II). Contrarily, if we apply one filter, we can have the first estimation in six weeks.

For practical purposes, we have seen that the application of the differential approach with one filter and setting the derivative threshold at 0.05 plants $(m^2 \, day)^{-1}$ gives the best result if we are interested in determining the week at which a weed control measure should be carried out, but if we are interested in the weed seedling population that is expected to emerge in the field, the application of two filters and setting a smaller threshold for the derivative (i.e., 0.03 plants $(m^2 \, day)^{-1}$) gives better estimations. Recall that the thresholds should be set according to the knowledge of the competition effect of the weed on the crop.

When we are following the emergence of weeds in crops such as winter cereals in which post-herbicide applications are carried out several months after sowing, weekly counts are appropriate. This interval has also proved to be appropriate for predicting emergence in irrigated crops like corn or citrus. However, crops with shorter growing seasons such as vegetables where crop-weed competition may start early might require more frequent counts to better predict timings and weed densities, particularly, considering that the use of one filter to smooth the data requires to have three counts completed.

The use of the differential form of any equation requires real-time data of emergences. This means that we have to count weed seedlings every time, usually every week, which can be very time-consuming. In this sense, the use of the integrated model only requires meteorological data which can be available from official weather stations or web-based weather records which is far more practical. However, in some regions weather stations can be very sparse, data cannot be available to the public or the emergence pattern of a given weed is not known in that particular region. In these cases, the use of the differential approach can help to manage weeds efficiently. Another advantage of this approach is that because we are using current data of the emergences on the calculations, any climatic factor that may affect the germination of the seeds that particular year (i.e.,: drought, wet weather, low temperatures) favoring the germination or delaying it, are reflected in the counts so the predictions will be more accurate than if we use an integrated model that has been developed under a given climatic conditions.

This flexibility is a characteristic of any differential model. We have tested the differential approach using the Gompertz function because it has been successfully used in its integrative form. Despite other weather-based sigmoidal functions that could be tested in their differential forms, such as Weibull, Richards, or logistic, Gompertz reproduces correctly the asymmetric dynamics of the weed emergence with a minimal number of parameters to be fitted. The build-up of the curve according to the function is an iterative process that uses the previous points plus adding new ones, so the accuracy of predicting the week of maximum emergence is only dependent on how closely the data follow the kinetics. In this sense, the more accurate the estimates of the thermal or hydrothermal time accumulated by the seeds, the better the predictions. It is also important to remark that the accuracy in predicting the maximum density of seedlings expected increases as more counts become available.

One limitation of the differential approach that we have seen is to warn the maximum seedling emergence too early because of the deceleration or even the cessation of the emergences for a given period of time, before the end of the emergences. Some days later the emergence resumes. This behavior is reflected in the curve of the cumulative emergences as step-like discontinuities. However, this is due to the misestimation of the thermal or hydrothermal time currently accumulated by the seeds due to the above-mentioned complexity of the weather-seed interactions or due to the existence of distinct subpopulations that result in different flushes of emergences [10]. If we are following emergences using a differential approach, we must be aware of such false positives by having an idea of the hydrothermal time at which the species is expected to complete the emergence. Nowadays, the use and popularity of hyperspectral cameras to monitor soil moisture will increase the accuracy of the estimation of this kind of data at field level [33]. This will allow better estimations of the hydrothermal time accumulated by the seeds and an improvement in the accuracy of the predictions.

Furthermore, it is also important to remember that in order to optimize the efficacy of the decisions, additional factors should be considered such as the growth stage of the crop, because if it is too developed it can make the control measure difficult to execute, or the average growth stage of the weed, because if we wait until the date of the maximum emergence, some early emerged weeds may become too large for adequate control.

The differential approach has been shown to be accurate enough in estimating the maximum weed emergence expected in the field and the time at which this maximum emergence will be reached. Weed management is a complex task that requires the integration of

many factors, one of them being the knowledge of the emergence pattern of the weeds in order to optimize the timing of post-emergence control measures such as herbicide spraying or tillage. As no single integrated model has so far proven able to describe the emergence pattern of all plant species in all scenarios, the incorporation of the differential approach into the decision support systems provides more flexibility when using hydrothermal time models to forecast weed emergences. This fact can be crucial, particularly during the transitional periods in which an integrated model is being developed. By adding this tool to weed management programs, an increase in the sustainability of the agricultural systems due to a reduction of herbicide use and weed control costs is expected.

**Supplementary Materials:** The following supporting information can be downloaded at: https://www.mdpi.com/article/10.3390/agronomy12112896/s1, Table S1: Details of the climatic and soil conditions of the fields in which the emergences were recorded.; Table S2: Some examples of the calculations of the differential approach applied to the emergence of *Papaver rhoeas* at Igualada 2008.

**Author Contributions:** Conceptualization, D.L. and J.I.; methodology, C.P., D.L. and J.I.; software, C.P.; validation, J.I.; formal analysis, C.P. and D.L.; investigation, J.I., M.G.; data curation, J.I.; writing—original draft preparation, J.I.; writing—review and editing, J.I., C.P. and D.L.; supervision, J.I. All authors have read and agreed to the published version of the manuscript.

**Funding:** This research received no external funding.

**Data Availability Statement:** The data presented in this study are available on request from the corresponding author.

**Acknowledgments:** We thank José Dorado, Edite Sousa, Isabel Calha, José Luis González-Andújar and César Fernández-Quintanilla for sharing their weed emergence data of corn fields.

**Conflicts of Interest:** The authors declare no conflict of interest.

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
