# Peer review of "A New Approach for Timing Post-Emergence Weed Control Measures in Crops: The Use of the Differential Form of the Emergence Model"

_agronomy, doi:10.3390/agronomy12112896_

Round 1
Reviewer 1 Report
A very interesting approach has been presented and supported by the results. I found the paper very useful and has an impactful contribution to the literature. I have the following comment that may improve the paper.
1. Hyperspectral cameras are also widely used to predict soil moisture. Related papers can be reviewed to support the hydrothermal time models.
2. Would it make a sensible difference if the other functions like logistic or Weibull were used in terms of accuracy?
3. Equation numbers should be in line.
4. Equation number 3 is given to two different equations. It should be corrected.
5. For Table 2, the error for the specified parameter should be given.
Author Response
- Hyperspectral cameras are also widely used to predict soil moisture. Related papers can be reviewed to support the hydrothermal time models.
We thank the reviewer for this valuable information. We have added a comment in the Discussion section about the increasing use of hyperspectral cameras in determining the water content of plants and soils and how this fact can improve emergence estimations.
- 2. Would it make a sensible difference if the other functions like logistic or Weibull were used in terms of accuracy?
No, it wouldn’t. A comment about the no difference of using other functions instead of Gompertz has been added in the discussion section. The key point in getting a good prediction is not the sigmoidal function used (either in its differential or integrative form) but the existence of a good sigmoidal relationship between the observed emergence in the field and time. If this is not true, any function form will fail.
- Equation numbers should be in line.
We see all equation numbers in line.
- 4. Equation number 3 is given to two different equations. It should be corrected.
Corrected.
- 5. For Table 2, the error for the specified parameter should be given.
We have added the errors.
Reviewer 2 Report
The manuscript agronomy-1943720 is a very interesting one; however, it requires some changes to get published in the Journal.
1. The same message has been written twice (in lines 25 and 30). Please discard one statement (maybe line 30 is not required).
2. Lines 36-37 should be rewritten to make the sentence more meaningful.
3. In the Materials and Methods section, it should be better not to give a heading for background. In the case of the experimental data set, 39 emergence data sets were used. However, the names of the sites are only given in Table 1 where the country name and seasonal and environmental information are missing. Yes, the author has provided the information on the water regime; but still, some climatic and soil data sets are required to get a clear idea about the environment of weed emergence.
4. The authors used the terms like ‘weed emergence’ and ‘seedlings’ for several times. It might make confusion among the readers about the seedlings whether the seedlings of crops or weeds. It would be wise to clearly mention weed seedlings.
5. The procedure of weed sample collection is good. However; it would be better to mention the parameters of data collection.
6. Eq. 3 has already been mentioned in line 145. So, the serial number of the equation in line 203 should Eq. 4. and the next equation numbers will be changed consecutively.
7. Please rewrite the sentence in line 256.
8. The rationale for doing this excellent piece of work has not been written well in the introduction section. The justification for conducting research is not expectedly aligned with the title.
Author Response
- The same message has been written twice (in lines 25 and 30). Please discard one statement (maybe line 30 is not required).
Corrected. Line 30 has been deleted.
- Lines 36-37 should be rewritten to make the sentence more meaningful.
We have changed “weed management” by “weed control” in the sentence.
- In the Materials and Methods section, it should be better not to give a heading for background. In the case of the experimental data set, 39 emergence data sets were used. However, the names of the sites are only given in Table 1 where the country name and seasonal and environmental information are missing. Yes, the author has provided the information on the water regime; but still, some climatic and soil data sets are required to get a clear idea about the environment of weed emergence.
“Background” has been deleted according the suggestion of the reviewer. The climatic and soil data that the reviewer is demanding can be found in the published papers that are mentioned in the text: Dorado et al. (2009) and Izquierdo et al. (2009 and 2013). However, and in order to facilitate this information to the reader, we have added this information in the Suplementary material. We believe that this information is not relevant considering the objective of the paper (the presentation of a new methodology using already published data) and, if added to the Table, this one will be too large.
- The authors used the terms like ‘weed emergence’ and ‘seedlings’ for several times. It might make confusion among the readers about the seedlings whether the seedlings of crops or weeds. It would be wise to clearly mention weed seedlings.
Following the suggestion of the reviewer, we have added the word “weed” before “seedlings” in many sentences to make clearer that we are referring to weed seedlings and not the crop.
- The procedure of weed sample collection is good. However; it would be better to mention the parameters of data collection.
We don’t understand what the reviewer is referring to when saying “parameters”. In Materials and methods, we already give essential information about how the data was recorded in the fields. More information about how the data were collected can be found in the cited literature (Dorado et al. (2009) and Izquierdo et al. (2009 and 2013)).
For each weed species, the emergence was counted weekly in 20 randomly placed permanent quadrats of 30 cm x 40 cm to 50 cm x 50 cm depending on the field. After counted, weed seedlings were removed from the soil with minimum soil disturbance. Data loggers were placed in each field to record soil tem-perature hourly
- Eq. 3 has already been mentioned in line 145. So, the serial number of the equation in line 203 should Eq. 4. and the next equation numbers will be changed consecutively.
Thanks. This mistake has been corrected.
- Please rewrite the sentence in line 256.
Thanks. The sentence has been rewritten as “Farmers accept that ….”
- The rationale for doing this excellent piece of work has not been written well in the introduction section. The justification for conducting research is not expectedly aligned with the title.
We have added some new words to the title. With this new title, we have tried to provide as much information as possible and clearly indicate the content of the paper. We also try to attract the attention of the target readers effectively, so that they will want to access and read the whole document.
Reviewer 3 Report
The manuscript is well written and the topic is important. But there are some problems in presenting the results. It should be shown how the hydrothermal time is calculated? Where were the parameters of this model obtained for different plants? Was the hydrothermal time model obtained for each plant or did you obtain its parameters from sources? It is better to bring a table containing the parameters of the hydrothermal time model for each plant. I suggest that the hydrothermal time models fitted to the data should be shown in the supplementary data file.
In parts of the text, there are numbers that do not have units. Also, in some parts, the units provided seem to have problems. For example, in line 141, The unit of hydrothermal time is MPa oC day-1.
In parts of the text, you referred to “germination”, for example in line 146, but I think you mean “seedling emergence”. Please check it.
In Fig 1, fitting of a line with two point is not accurate. You need at least three point to fit a line. The unit of emergence rate is not “oC -1”. The coefficients of the equations are almost the same for all weeks. Why not stack all the data and get one model for all weeks?
Author Response
- The manuscript is well written and the topic is important. But there are some problems in presenting the results. It should be shown how the hydrothermal time is calculated? Where were the parameters of this model obtained for different plants? Was the hydrothermal time model obtained for each plant or did you obtain its parameters from sources? It is better to bring a table containing the parameters of the hydrothermal time model for each plant. I suggest that the hydrothermal time models fitted to the data should be shown in the supplementary data file.
It is not the objective of this paper to show how the thermal (TT) or hydrothermal time (HTD) were calculated for each data sets. The TT and HDT were already calculated and set (see Dorado et al; Izquierdo et al 20134). What we present here is a new approach to relating this time (TT or HDT) to the emergences observed in the field. Usually this is done by using the integrative form of the functions. However, we suggest to use the differential form of the function. In order to check the feasibility of this approach, we implemented this approach in 39 datasets and explained the results. Furthermore, we graphically showed the implementation of the protocol in one of them (Igualada).
The 39 emergence datasets selected were originally related to the thermal or hydrothermal time with a function in its integrative form. These functions can be found in the above-mentioned references. And the estimation of the TT and HTD accumulated by the seeds was done in these ways:
1) The TT as TT=(T-Tb) where T is the daily soil temperature and Tb is the threshold T that will just prevent germination of the seeds (Gummerson 1986; Bradford 2002).
2) The HDT as the thermal time plus a correction based on a daily water balance of the soil (for more details see Izquierdo et al. 2013).
- In parts of the text, there are numbers that do not have units. Also, in some parts, the units provided seem to have problems. For example, in line 141, the unit of hydrothermal time is MPa oC day-1.
We have added units to the thresholds and corrected the units of the hydrothermal time.
- In parts of the text, you referred to “germination”, for example in line 146, but I think you mean “seedling emergence”. Please check it.
We have corrected and changed all words that may lead to confusion.
- In Fig 1, fitting of a line with two point is not accurate. You need at least three point to fit a line. The unit of emergence rate is not “oC -1”. The coefficients of the equations are almost the same for all weeks. Why not stack all the data and get one model for all weeks?
Although mathematically two points are enough to fit a line, we kindly explain the objective of this figure. What we want to show with this figure is graphically represent the procedure that we have followed with the iterations: how the coefficients vary when adding one more datum point every week. Each graph represents one more week of data and we show how the fitting varies. Obviously, the variation in relation to the previous graph (=previous week) is small because only one point, the number of new seedlings of that week, is added each time. The procedure is repeated till the derivative (shown in Table 2) is below 0.03 plants day (m2 MPa)-1. By keeping the graph of each week separately, we can better see the variation of the coefficients and how the procedure works.